# Lactic Acid Bacteria Improve the Photoprotective Effect via MAPK/AP-1/MMP Signaling Pathway on Skin Fibroblasts

**DOI:** 10.3390/microorganisms10122481

**Published:** 2022-12-15

**Authors:** Jeong-Yong Park, Ji Yeon Lee, YongGyeong Kim, Chang-Ho Kang

**Affiliations:** MEDIOGEN, Co., Ltd., Biovalley 1-ro, Jecheon-si 27159, Republic of Korea

**Keywords:** lactic acid bacteria, ultraviolet B, matrix metalloproteinases, activator protein-1, collagen

## Abstract

Ultraviolet B (UVB) exposure causes a breakdown of collagen, oxidative stress, and inflammation. UVB activates mitogen-activated protein kinase (MAPK), activator protein-1 (AP-1), and matrix metalloproteinases (MMPs). In this study, we evaluated 2,2′-azino-bis-(3-ethylbenzothiazoline-6-sulfonic acid) (ABTS^+^) radical scavenging activity and the photoprotective effect of lactic acid bacteria LAB strains, including *Lactobacillus*, *Bifidobacterium*, and *Streptococcus* genera in UVB-exposed skin fibroblasts. Nine LAB strains displayed antioxidant activity by regulating superoxide dismutase in UVB-exposed skin fibroblasts. Four LAB strains (MG4684, MG5368, MG4511, and MG5140) recovered type I procollagen level by inhibiting MMPs, MAPK, and AP-1 protein expression. Additionally, these four strains reduced the expression of proinflammatory cytokines by inhibiting oxidative stress. Therefore, *L. fermentum* MG4684, MG5368, *L. rhamnosus* MG4511, and *S. thermophilus* MG5140 are potentially photoprotective.

## 1. Introduction

Skin is composed of epidermis and dermis and is the first body part to be stimulated by external factors [1]. The most characteristic change that occurs with skin aging is a change in the extracellular matrix (ECM) [2]. Fibroblasts of the dermis display a decreased ability to form a matrix as they age. This decrease leads to the induction of wrinkles in the skin by increasing the thickness and decreasing the elasticity [3]. In the ECM, collagen is an important insoluble fiber synthesized mainly from fibroblasts present in the dermis; type 1 collagen is the most abundant subtype of collagen [4,5]. Collagen, which is closely related to skin elasticity, plays an important role in maintaining the shape of the skin by forming the structural skeleton of the skin [6,7]. Therefore, collagen degradation is a prominent feature of skin aging, and photoaging is one of the main mechanisms leading to collagen degradation [8,9,10].

Exposure to an appropriate amount of ultraviolet (UV) radiation stimulates the synthesis of vitamin D. However, UVB can detrimentally affect the skin [11,12]. The UV dose reaching the Earth’s surface has increased over the past 20 years. This has caused diverse biological responses in human skin, including aging, cancer, inflammation, and severe disease [13,14]. UVB radiation activates the phosphorylation of mitogen-activated protein kinase (MAPK), which affects the activator protein-1 (AP-1) signaling cascade [15]. Activated AP-1 induces matrix metalloproteinases (MMPs). The resulting interruption of procollagen type 1 synthesis results in the degradation of collagen and elastin, which are major components of the ECM [16]. In the dermis, damage to type 1 procollagen depends on the expression of MMP-1, which is a member of the MMP family that specifically degrades type 1 collagen [17]. Exposure to UVB induces inflammatory cytokines such as interleukin (IL)-6 and IL-1β, resulting in the overexpression of MMPs [18]. In skin aging, the expression of MMPs and collagen degradation are important mechanisms. Therefore, it is necessary to develop a safe dietary material that can prevent collagen degradation by UVB in the skin.

Lactic acid bacteria (LAB) attach to intestinal epithelial cells and remove harmful microorganisms, having beneficial biological effects [19]. Although many studies have addressed the regulation of intestinal health by LAB, a recent study reported that the intake of LAB is helpful in alleviating various diseases [20]. *Lactiplantibacillus plantarum*, *Bifidobacterium longum*, *Bi. breve*, and *Bi. bifidum* reportedly inhibit UVB-induced photoaging [21,22]. Although the skin photoprotective effect of various LAB strains have been reported, studies on some species are insufficient. Therefore, we evaluated the photoprotective effect of diverse LAB strains in UVB-exposed skin fibroblasts.

## 2. Materials and Methods

### 2.1. Preparation of Cell-Free Supernatant (CFS) from Lactobacillus, Streptococcus, and Bifidobacterium Strains

*L. plantarum* MG5023, MG5372, MG5302, and MG5433; *Limosilactobacillus fermentum* MG4231, MG4254, MG4635, MG4684, MG4698, MG4717, MG4721, MG4723, MG5368, and MG5389; *Limosilactobacillus reuteri* MG4658, MG5460, and MG5462; *Lacticaseibacillus rhamnosus* MG4511, MG5437, and MG5511; *Ligilactobacillus salivarius* MG242, MG4664, and MG4679; *S. thermophilus* MG5140, MG5475, and MG5481; *Bi. longum* MG723 and MG4669; and *Bi. bifidum* MG731 were obtained from MEDIOGEN (Jecheon, Republic of Korea). All LAB strains were cultured in MRS broth (BD Bioscience, Franklin Lakes, NJ, USA) at 37 °C. After 18 h, the CFS was obtained by centrifugation (4000× *g* for 15 min at 4 °C). The supernatants were filtered and sterilized using a 0.22 μm polytetrafluoroethylene membrane filter (ADVANTEC, Tokyo, Japan). To identify all LAB strains, 16S rRNA gene sequencing was confirmed (SolGent Co., Ltd., Daejeon, Republic of Korea). The DNA sequences were registered in the National Center for Biotechnology Information (NCBI) database using the Basic Local Alignment Search Tool (Table 1).

### 2.2. 2,2′-Azino-Bis-(3-ethylbenzothiazoline-6-sulfonic Acid) (ABTS^+^) Radical Scavenging Activity

ABTS^+^ radical scavenging activity was measured as previously described, with some modification [23]. ABTS^+^ solution was prepared using a 7 mM ABTS tablet (Sigma-Aldrich, St. Louis, MO, USA) with 2.45 mM potassium persulfate in distilled water for 4 h at 4 °C. All LAB strains were collected by centrifugation (4000× *g*, 15 min at 4 °C). Each pellet was resuspended in phosphate-buffered saline (PBS) at 1 × 10^8^ CFU/mL. Each LAB suspension (300 µL) and 600 µL of ABTS^+^ solution was mixed and incubated for 1 h at 37 °C. Absorbance was measured using a microplate reader at 734 nm. The results were calculated according to the following equation:ABTS^+^ scavenging activity (%) = (1 − A_treatment_/A_control_) × 100
where A_treatment_ is the absorbance value of the ABTS^+^ solution with sample treatment, and A_control_ is the absorbance value of the ABTS^+^ solution with PBS.

### 2.3. Cell Culture and UVB Treatment

Hs68 cells (ATCC, Manassas, VA, USA) were grown in DMEM (Gibco, Grand Island, NY, USA) supplemented with 10% fetal bovine serum (Gibco) and 1% penicillin–streptomycin (P/S, Gibco) at 37 °C in 5% CO_2_. UVB exposure to cells was measured according to a previously described method [24]. Hs68 cells were seeded at 1 × 10^5^ cells/mL in the culture area. After 24 h, the cells were treated with 10% CFS of LAB in the culture media for 24 h. The cells were washed once with PBS (Welgene, Daegu, Republic of Korea) and exposed to UVB (30 mJ/cm^2^) using a UV crosslinker (PCL-1000; BoTeck, Gunpo, Republic of Korea) without the culture plate cover. After UVB exposure, the cells were replaced with fresh culture medium and treated with CFS for an additional 24 h.

### 2.4. Cytotoxicity

Hs68 cells were treated with 10% CFS of LAB for 24 h with or without UVB irradiation. Cytotoxicity was investigated using the 3-(4,5-dimethylthiazol-2-yl)-2,5-diphenyltetrazolium bromide (MTT) assay [24]. MTT solution (0.2 mg/mL, Sigma-Aldrich) was added for 2 h. The formazan product was dissolved in dimethyl sulfoxide, and the absorbance was measured using an EPOCH2 microplate reader (BioTek, Winooski, VT, USA) at 550 nm.

### 2.5. Enzyme-Linked Immunosorbent Assay (ELISA)

Cells were pretreated with CFS and exposure with UVB in the same manner described in Section 2.3. Cell culture medium was collected, and superoxide dismutase (SOD) 1 (Abcam, Cambridge, MA, USA) and Type I procollagen (TaKaRa Bio, Shiga, Japan) were quantified following the manufacturer’s instructions using a microplate reader.

### 2.6. Quantitative Real-Time Polymerase Chain Reaction (qRT-PCR)

Cells were pretreated with CFS and exposure with UVB in the same manner described in Section 2.3. Total RNA from Hs68 cells was isolated using NucleoZOL reagent (MACHEREY-NAGEL, Dueren, Germany) according to the manufacturer’s instructions. RNA was synthesized using a cDNA reverse-transcriptase premix (iNtron Biotechnology, Seongnam-si, Republic of Korea). Gene expression was determined using amfiSure qGreen qPCR Master Mix (GenDEPOT, Katy, TX, USA). qRT-PCR was performed using a CFX Connect Real-Time PCR Detection System (Bio-Rad, Hercules, CA, USA). Target genes were normalized to that of glyceraldehyde 3-phosphate dehydrogenase (GAPDH) and analyzed by the 2^−∆∆CT^ method. The PCR primers used for this analysis are listed in Table 2.

### 2.7. Western Blot

Cells were treated with CFS and exposure with UVB in the same manner described in Section 2.3. For Western blot analysis, after UVB irradiation, Hs68 cells were treated with 10% CFS of LAB for 1–48 h. Whole cell lysates were prepared in RIPA buffer (GenDEPOT) containing protease and phosphatase inhibitor cocktail (GenDEPOT). Cell lysates with equal amounts of total protein were determined using a Bradford assay (GenDEPOT). The proteins were mixed with 5× sample buffer (iNtron Biotechnology) following the manufacturer’s instructions. Western blotting was performed by 10% sodium dodecyl sulfate-polyacrylamide gel electrophoresis (SDS-PAGE), followed by protein transfer to polyvinylidene fluoride (PVDF) membranes (0.45 μm; Millipore, Darmstadt, Germany). Each membrane was blocked for 5 min with fast blocking buffer (Biomax, Seoul, Republic of Korea). Then, the membrane was washed three times with Tris-buffered saline-Tween (TBS-T) and incubated with primary antibody (1:1000 dilution) containing 5% bovine serum albumin overnight at 4 °C. The membrane was washed again three times and treated with secondary antibody (1:5000 dilution) for 1 h at room temperature. The primary antibodies (β-actin, MMP-1, -3, p-cFos, p-cJun, cFos, and cJun mouse monoclonal or p-extracellular signal-regulated kinase (ERK), p-c-Jun, N-terminal kinase (JNK), p-p38, ERK, JNK, and p38 rabbit monoclonal) and horseradish peroxidase (HRP)-linked secondary antibodies (anti-rabbit and anti-mouse IgG) were purchased from Santa Cruz Biotechnology (Dallas, TX, USA), Cell Signaling Technology (Beverly, MA, USA), and GenDEPOT. Protein bands were visualized using a LuminoGraph III Lite (ATTO, Tokyo, Japan) with West-Q Femto Clean ECL solution (GenDEPOT). Quantitative analysis was conducted using ImageJ software (version 1.52a for Windows; NIH, Rockville, MD, USA).

### 2.8. Statistical Analysis

All experimental results are expressed as mean ± standard error of the mean (SEM) of three independent measurements. Statistical analysis was performed by Duncan’s multiple range test using SPSS, ver. 22.0 for Windows (IBM, Armonk, NY, USA). Statistical significance was set at *p* < 0.05.

## 3. Results

### 3.1. Antioxidant Activity of LAB Strains by ABTS^+^ Radical Scavenging Activity

To estimate the antioxidant activity of LAB, an ABTS^+^ radical scavenging assay was performed. The ABTS^+^ radical scavenging activity of the LAB strains ranged from 23.3 to 93.1%. Based on the assay results, 19 LAB strains with more than 60% radical scavenging ability were selected (Table 3). The protective effects of these strains on UVB-exposed Hs68 cells were confirmed.

### 3.2. CFS of LAB Improves SOD Enzyme and Type I Procollagen Levels in UVB-Exposed Hs68 Cells

Hs68 cell viability was >100% for CFS levels of up to 10% (Figure 1A). Therefore, subsequent experiments used 10% CFS from the LAB. The cell viability was significantly increased by CFS treatment of all LAB strains in UVB-exposed Hs68 cells (Figure 1B).

We measured the SOD levels in UVB-exposed Hs68 cells. SOD levels were significantly reduced by UVB exposure (0.22-fold of control), while all LAB strains, except MG4231, MG4635, MG4723, MG5389, MG5462, MG4664, MG4679, MG723, and MG731 significantly restored SOD levels (2.44 to 5.58-fold of UVB-control) (Figure 1C). We also evaluated the effects on type I procollagen production in UVB-exposed Hs68 cells. UVB exposure significantly reduced procollagen levels (0.39-fold of control). In LAB strains, except MG5023, MG4254, and MG723, 10% CFS treatment significantly improved the type I procollagen level compared with that in the UVB-control (1.15 to 2.47-fold of UVB-control) (Figure 1D).

Based on the results of SOD and type I procollagen levels on UVB-exposed Hs68 cells, six LAB strains (*L. fermentum* MG4684, MG4721, MG5368; *L. rhamnosus* MG4511, MG242; and *S. thermophilus* MG5140) were selected for the determination of type I procollagen levels in Hs68 cells. The type I procollagen levels in Hs68 cells not exposed to UVB exposure, *L. fermentum* MG4684 (1.53-fold of control), MG4721 (1.51-fold of control), MG5368 (1.60-fold of control), *L. rhamnosus* MG4511 (1.72-fold of control), *L. salivarius* MG242 (1.34-fold of control), and *S. thermophilus* MG5140 (1.34-fold of control) were confirmed (Figure 1E).

### 3.3. CFS of LAB Inhibits mRNA Expression of Proinflammatory Cytokines in UVB-Exposed Hs68 Cells

The effect of selected LAB strains on proinflammatory cytokine mRNA expression in UVB-exposed Hs68 cells was evaluated. The mRNA expression of *IL-6* was significantly increased by UVB exposure compared with that in the control, while the CFS of LAB strains significantly reduced *IL-6* mRNA expression (Figure 2). *IL-6* mRNA expression in all LAB strains was significantly diminished from 0.20- to 0.35-fold that of the UVB-control. In addition, the *IL-1β* mRNA expression was increased by UVB exposure compared with that in the control (Figure 2). In all LAB strains, *IL-1β* mRNA expression was reduced (0.28- to 0.68-fold of that of the UVB-control), except for *L. fermentum* MG4684. Only *S. thermophilus* MG5140 significantly inhibited *IL-1β* mRNA expression (0.28-fold of that of the UVB-control).

### 3.4. CFS of LAB Inhibits mRNA and Protein Expression of MMP-1 and MMP-3 in UVB-Exposed Hs68 Cells

To determine the effect of CFS from LAB strains on the MMP-1 and MMP-3 expression, qRT-PCR analysis and Western blot were performed. The mRNA expressions of *MMP-1* and *MMP-3* were significantly increased in UVB-exposed Hs68 cells compared with those of cells in the control group (Figure 3A). The mRNA expression of *MMP-1* was significantly reduced in all LAB strains in UVB-exposed Hs68 cells (0.02- to 0.41-fold of UVB-control). Additionally, the mRNA expression of *MMP-3* was decreased by treatment with LAB strains (0.47 to 0.76-fold of UVB-control), except for *L. salivarius* MG242 and *S. thermophilus* MG5140.

Based on the results of the mRNA expression of proinflammatory cytokines and MMPs on UVB-exposed Hs68 cells, four LAB strains (*L. fermentum* MG4684 and MG5368, *L. rhamnosus* MG4511, and *S. thermophilus* MG5140) displayed markedly inhibited mRNA expression of proinflammatory cytokines and MMPs. Thus, we assessed the effect of these four LAB strains on MMP-1 and MMP-3 protein expression in UVB-exposed Hs68 cells. The protein expressions of MMP-1 and MMP-3 were significantly increased with UVB exposure compared with those in the control (Figure 3B). The protein expressions of MMP-1 and MMP-3 were decreased by LAB strain treatment of UVB-exposed Hs68 cells compared with those of the UVB-control (MMP-1, 0.43- to 0.65-fold of UVB-control; and MMP-3, 0.71- to 0.80-fold of UVB-control).

### 3.5. CFS of LAB Inhibit Protein Expression of MAPK and AP-1 Signaling Pathway in UVB-Exposed Hs68 Cells

To investigate the effects of the CFS from LAB strains on the MAPK and AP-1 protein expression, Western blotting was performed. The protein expressions of p-ERK, p-JNK, and p-p38 were significantly increased by UVB exposure. In contrast, treatment with LAB strains reduced p-ERK (0.59- to 0.86-fold of UVB-control), p-JNK (0.86-fold of UVB-control), and p-p38 (0.16- to 1.00-fold of UVB-control, Figure 4A). UVB exposure also induced the protein expressions of c-Fos and cJun phosphorylation (Figure 4B). cFos expression was reduced in MG4684 (0.67-fold of UVB-control) and MG5368 (0.83-fold of UVB-control), and that of cJun was reduced in MG4684 (0.73-fold of UVB-control), MG5368 (0.81-fold of UVB-control), MG4511 (0.54-fold of UVB-control), and MG5140 (0.56-fold of UVB-control).

## 4. Discussion

UV radiation activates several pathways and causes damage to the skin, resulting in photoaging. UVA and UVB rays can reach the Earth’s surface and affect the human body; UVB is reportedly a major factor in photoaging [25]. In addition, the amount of UVB reaching the Earth’s surface is increasing due to environmental pollution [14]. Continuous UVB exposure causes skin effects that include photoaging, inflammation, and reduction in collagen, which lead to decreased elasticity and wrinkles [26]. Therefore, agents that protect skin against UVB are required. The genera most commonly found in the gut, such as *Lactobacillus*, *Bifidobacterium*, and *Streptococcus*, are beneficial for the skin [21]. *L. plantarum* HY7714 has confirmed antiwrinkle effects on skin cells, in vivo models, and clinical studies [19,20,27]. In one clinical trial, *S. thermophilus* TCI633 improved wrinkles and elasticity [28]. *L. casei* 327 improved skin barrier in the same clinical study [29]. *Bi. breve* B-3 reduced skin thickness in UVB-exposed HR-1 hairless mice [30]. Probiotics can affect the composition of the gut microbiome, which can affect other organs by altering the immune system through the production of metabolites [31]. Probiotic metabolites in the CFS, including lipoteichoic acid, lactic acid, and acetic acid, improve skin and its barrier function [32]. In the present study, the photoprotective effect of CFS from LAB was confirmed in UVB-exposed Hs68 cells.

UVB stimulates the production of oxidative stress, which leads to skin lesions and results in accelerated aging [6]. Oxidative stress exists in diverse forms, including free and superoxide anion radicals [33]. Antioxidants scavenge free radicals that negatively impact organisms [34]. ABTS^+^ radical scavenging activity is a rapid and simple assay that evaluates the ability to eliminate free radicals used for screening to evaluate antioxidant potential [35]. SOD plays an important role in catalyzing the scavenging of oxidative stress free radicals [36]. SOD scavenges radicals by converting the superoxide anion radical to H_2_O_2_ [37]. The skin has antioxidant enzymes that include SOD [38]. In the present study, the free radical scavenging potential of the LAB strains was confirmed through an ABTS^+^ radical scavenging activity, and the inhibition of oxidative stress in UVB-induced skin fibroblast was confirmed in nine LAB strains through the SOD activity. Oxidative stress causes the overexpression of the proinflammatory cytokines IL-6 and IL-1β via MAPK activation [18]. IL-6 and IL-1β mediate skin inflammation by activating inflammation-related signaling pathways [39]. Suppression of proinflammatory cytokines is an effective means of preventing inflammation in the skin [18]. In the present study, *L. fermentum* MG4684, MG4721, MG5368, *L. rhamnosus* MG4511, *L. salivarius* MG242, and *S. thermophilus* MG5140 remarkably diminished the mRNA expressions of *IL-6* and *IL-1β*. The six LAB strains diminished the inflammatory response by suppressing oxidative stress.

UVB promotes MAPK, which acts as a mediator intracellular signaling factor, including AP-1 [26]. AP-1, composed of subunits of cFos and cJun, is a representative factor regulating gene expression [40]. In the present study, *L. fermentum* MG4684 and MG5368, *L. rhamnosus* MG4511, and *S. thermophilus* MG5140 diminished the protein expressions of p-ERK, p-JNK, and p-p38. In addition, *L. fermentum* MG4684, *L. rhamnosus* MG4511, and *S. thermophilus* MG5140 inhibited the expression of p-cJun, but they did not affect p-c-Fos expression. Activated AP-1 moves from the cytoplasm into the nucleus to promote the expression of MMPs and inhibits the synthesis of procollagen, a precursor of collagen, inducing photoaging of the dermal layer [41]. Procollagen is synthesized in dermal fibroblasts and is secreted from the extracellular space, then combine with each other to form type 1 collagen [8]. UVB exposure accelerates the breakdown of collagen, which is a major component of the ECM [42]. UVB induces the expression of MMPs by activating the expressions of the MAPK and AP-1 signaling pathways [15]. MMPs, including MMP-1 and MMP-3, mainly degrade type I collagen, the most abundant component in dermal cells [43]. MMP-1 is the most important factor in collagen breakdown because it degrades collagen into two fragments, which initiates collagen breakdown [44]. We found that *L. fermentum* MG5368 and MG4684, *L. rhamnosus* MG4511, and *S. thermophilus* MG5140 improved type I procollagen regulating MMPs in UVB-exposed Hs68 cells by regulating MAPK and AP-1.

The findings confirm that *L. fermentum* MG4684 and MG5368, *L. rhamnosus* MG4511, and *S. thermophilus* MG5140 improved type I procollagen level by regulating the MMPs/MAPK/AP-1 signaling pathways in UVB-exposed skin fibroblasts. These four LAB strains also suppressed the expressions of proinflammatory cytokines by suppressing UVB-induced oxidative stress. Although the effect of LAB has been confirmed in cell models, animal models and clinical studies are needed to determine the skin protective effect against UVB exposure.

## 5. Conclusions

In summary, treatment with LAB increased SOD and type I procollagen levels and reduced the mRNA and protein expressions of MAPK, AP-1, MMPs, and inflammatory cytokines in UVB-exposed Hs68 cells. LAB can regulate the expression of MMPs through the MAPK/AP-1 signaling pathway. The findings implicate *L. fermentum* MG4684 and MG5368, *L. rhamnosus* MG4511, and *S. thermophilus* MG5140 as potential probiotics and functional food for the prevention of skin aging.

## Figures and Tables

**Figure 1 microorganisms-10-02481-f001:**
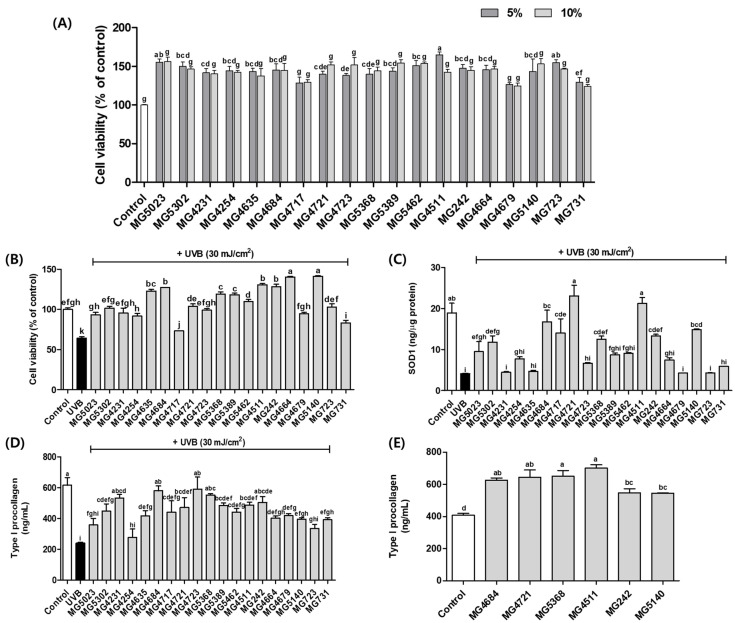
Effect of CFS from LAB strains on cell viability without UVB (**A**) or with UVB (**B**), SOD1 level (**C**), and type I procollagen level without UVB (**D**) or type I procollagen level with UVB (**E**). Data represent the mean ± SEM (*n* = 3). Different letters indicate significant difference between means at *p* < 0.05 by Duncan’s multiple range test.

**Figure 2 microorganisms-10-02481-f002:**
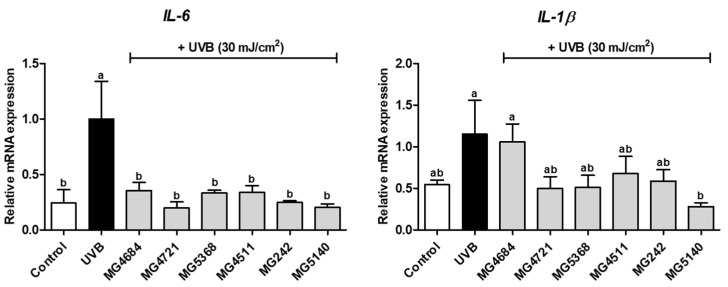
Effect of CFS from LAB strains on mRNA expression of *IL-6* and *IL-1β*. Data represent the mean ± SEM (*n* = 3). Different letters indicate significant difference between means at *p* < 0.05 by Duncan’s multiple range test.

**Figure 3 microorganisms-10-02481-f003:**
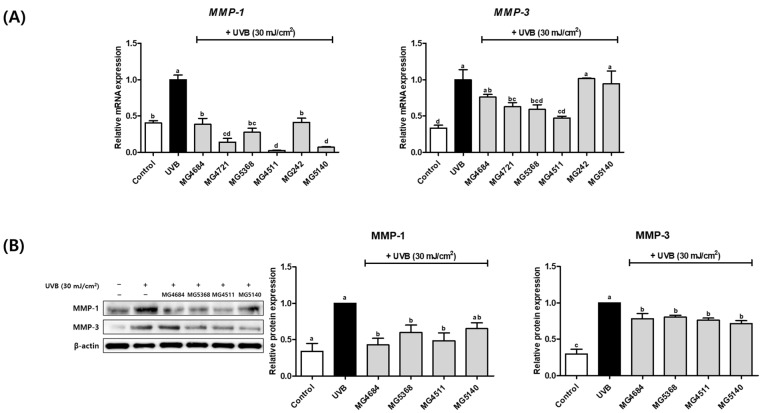
Effect of CFS from LAB strains on *MMP-1* and *MMP-3* mRNA expression (**A**) and MMP-1 and -3 protein expression (**B**) in UVB-exposed Hs68 cells. The results are presented as the mean ± SEM (*n* = 3). Different letters indicate significant difference between means at *p* < 0.05 by Duncan’s multiple range test.

**Figure 4 microorganisms-10-02481-f004:**
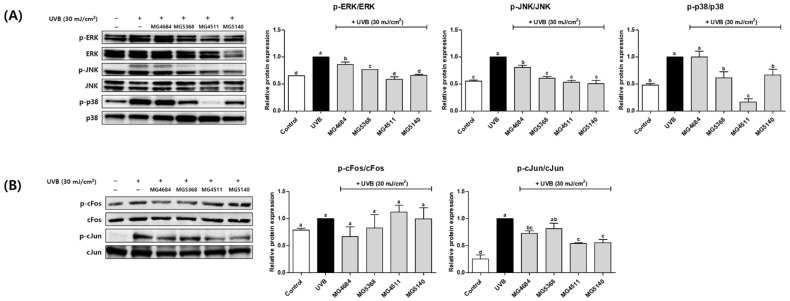
Effect of CFS from LAB strains on the protein expression of MAPK (**A**) and AP-1 (**B**) in UVB-exposed Hs68 cells. The results are presented as the mean ± SEM (*n* = 3). Different letters indicate significant difference between means at *p* < 0.05 by Duncan’s multiple range test.

**Table 1 microorganisms-10-02481-t001:** Accession number and origin of LAB used in this study.

LAB	Strain	NCBI Accession Number	Origin
*Lactiplantibacillus plantarum*	MG5023	OP102478.1	Food
	MG5372	ON631286.1	
	MG5302	MN833016.1	Fermented food
	MG5433	ON631843.1	
*Limosilactobacillus fermentum*	MG4231	MW947163.1	Human
	MG4254	MW947155.1	
	MG4635	ON631318.1	
	MG4684	OP035518.1	
	MG4698	OP077101.1	
	MG4717	OP035525.1	
	MG4721	OP035529.1	
	MG4723	OP035531.1	
	MG5368	ON631268.1	Food
	MG5389	ON668175.1	
*Limosilactobacillus reuteri*	MG4658	OP077269.1	Human
	MG5460	ON619506.1	Food
	MG5462	ON619508.1	
*Lacticaseibacillus rhamnosus*	MG4511	MN368054.1	Human
	MG5437	ON631847.1	Food
	MG5511	ON631745.1	
*Ligilactobacillus salivarius*	MG242	ON631745.1	Human
	MG4664	OP077274.1	
	MG4679	OP035513.1	
*Streptococcus thermophilus*	MG5140	MN055727.1	Fermented food
	MG5475	ON619521.1	
	MG5481	ON619527.1	
*Bifidobacterium longum*	MG723	MN055725.1	Human
	MG4669	OP035506.1	
*Bifidobacterium bifidum*	MG731	MN055710.1	Human

**Table 2 microorganisms-10-02481-t002:** Primer sequences for qRT-PCR.

Gene	Primer Sequence (5′ → 3′)	
*MMP-1*	F ^1^	GCATATCGATGCTGCTCTTTC
	R ^2^	GATAACCTGGATCCATAGATCGTT
*MMP-3*	F	CAAAACATATTTCTTTGTAGAGGACAA
	R	TTCAGCTATTTGCTTGGGAAA
*IL-6*	F	AAGCCAGAGCTGTGCAGATGAGTA
	R	TGTCCTGCAGCCACTGGTTC
*IL-1β*	F	AGAAGTACCTGAGCTCGCCA
	R	CCTGAAGCCCTTGCTGTAGT
*GAPDH*	F	ACCCACTCCTCCACCTTTG
	R	ACCCACTCCTCCACCTTTG

^1^ F, forward; ^2^ R, reverse; *MMP*, matrix metalloproteinase; *IL*, interleukin; *GAPDH*, glyceraldehyde 3-phosphate dehydrogenase.

**Table 3 microorganisms-10-02481-t003:** Antioxidant activity of LAB based on ABTS^+^ radical scavenging activity.

Species	Strain	Inhibition Rate (%)
	(1 × 10^8^ CFU/mL)	
*Lactiplantibacillus plantarum*	MG5023	69.9 ± 3.0 ^fg^
	MG5372	23.3 ± 0.6 ^o^
	MG5302	64.6 ± 3.4 ^i^
	MG5433	23.3 ± 1.0 ^o^
*Limosilactobacillus fermentum*	MG4231	75.7 ± 0.1 ^e^
	MG4254	64.6 ± 2.0 ^i^
	MG4684	68.6 ± 1.3 ^gh^
	MG4698	32.5 ± 0.2 ^kl^
	MG4717	64.7 ± 2.4 ^i^
	MG4721	77.2 ± 1.9 ^de^
	MG4723	86.9 ± 1.5 ^c^
	MG4635	93.1 ± 0.4 ^a^
	MG5368	72.1 ± 2.6 ^f^
	MG5389	79.2 ± 1.8 ^d^
*Limosilactobacillus reuteri*	MG4658	41.9 ± 0.9 ^j^
	MG5460	35.0 ± 0.6 ^k^
	MG5462	64.7 ± 2.4 ^i^
*Lacticaseibacillus rhamnosus*	MG4511	64.9 ± 1.1 ^i^
	MG5437	27.0 ± 1.5 ^n^
	MG5511	27.1 ± 0.7^n^
*Ligilactobacillus salivarius*	MG242	92.8 ± 0.7 ^a^
	MG4664	92.7 ± 0.8 ^a^
	MG4679	78.3 ± 0.3 ^de^
*Streptococcus thermophilus*	MG5140	65.9 ± 2.3 ^hi^
	MG5475	30.6 ± 3.9 ^lm^
	MG5481	28.9 ± 2.0 ^mn^
*Bifidobacterium longum*	MG723	89.2 ± 0.3 ^bc^
	MG4669	27.9 ± 0.1 ^mn^
*Bifidobacterium bifidum*	MG731	90.6 ± 0.2 ^ab^

All data are presented as the mean ± SEM (*n* = 3). Different letters indicate significant difference between means at *p* < 0.05 by Duncan’s multiple range test.

## Data Availability

Not applicable.

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
