# Peer review of "Lactic Acid Bacteria Improve the Photoprotective Effect via MAPK/AP-1/MMP Signaling Pathway on Skin Fibroblasts"

_microorganisms, 2022, doi:10.3390/microorganisms10122481_

Round 1

Reviewer 1 Report

Please verify the attached file and provide corrections as indicated.

Author Response

Reviewer 1

Thanks for your attentive comments.

The manuscript was revised based on the comments in the review, with changes marked in red.

Below is a summary of the answers to each comment.

<GENERAL COMMENTS>

Point 1: I disagree with this keyword.

Response: Thanks for your comment. According to the reviewer's comments, we modified the keywords.

Point 2: Same idea repeated. Please rewrite. (Lines 28-30)

Response: Thank you for your comments. We modified the sentences in lines 29-31.

Point 3: were purchased/achieved from (line 66)

Response 3: Thanks for your comments. We modified it to "achieved" in line 63.

Point 4: Indication of the meaning of superscripts is needed (line 153-154)

Response 4: Thanks for your comments. We added the meaning of the superscripts in Table 3 and Figure legends (1-4).

Point 5: Figure 1 need to improve these graphs. It is difficult to read.

Response 5: Thanks for your comments. According to the reviewer's comments, we modified Figure 1 to clearly represent the results.

Point 6: Change suppressed to diminished (line 261, 262, 267).

Response 6: Thanks for your comments. According to the reviewer's comments, we changed “suppressed” to “diminished” in lines 271, 272, and 276.

Reviewer 2 Report

The MS submitted by Part et al. claims that cell-free supernatants (CFS) of lactic acid bacteria strains have antioxidant activity through increasing expression of SOD1 and type I procollagen, and inhibiting expression of inflammatory cytokines and MMPs induced by UVB. The findings are novel in the sense that lactic acid bacteria release a molecule(s) that renders protection to Hs68 cells to UVB. The experiments performed seem to be appropriate, however, the procedures and rationale are not adequately described, which makes it hard to understand the mechanism involved in the protective effects. Below are more specific comments on the issues.

1. It is not clear when and why UVB and CSF were treated. For antioxidant activity analysis, CSF was directly mixed with ABTS reagent. For ELISA assays, cells were first treated with UVB for 24 h and then treated with CSF for 24h (line 103 to 104); whereas for general UVB treatments, cells were first treated with CFS for 24 h and then exposed to UVB (line 92 to 95). Please provide the rationale for treating UVB and CSF in different sequences in the text. It is presumed that cells were treated with CSF and then treated with UVB. It should be revealed that when qPCR and Western blot samples were taken after UVB treatment.

2. It is not clear why the radical scavenging activity of CSF is relevant to the subsequent experiments exploring the UVB protective effects of CSF. Strains of LAB were selected based on these activities for the following experiments, but CSF was not presented when UVB was exposed in these experiments. Please provide how the direct scavenging activity is involved in the UVB protective effects. Also, scavenging activity in the cytosol, where free radicals are made, is important. Please explain how direct scavenging effects in the extracellular are relevant in cells.

3. Based on Fig. 1, CSFs positively affect cell proliferation. UVB will likely induce cell death which could be inhibited in CSF-treated cells. Since SOD1 and type I procollagen experiments in Fig. 1B and 1C were done in cell culture media, cell viabilities of CSF-treated cells in response to UVB exposure will likely affect the results. Therefore, cell viability assays of these cells should be provided.

4. SOD1 is a cytosolic protein. Please provide the rationale for analyzing the levels in the cell culture media. Is it due to increased secretion or expression of these molecules? Please show if CSF induces mRNA or protein expression in Hs68 cells.

5. Fig 2-4 clearly show that CSF has anti-inflammatory and MAPK inhibitory effects. However, since CSF was not present when UVB was exposed, it is not clear if these effects are due to residual radial scavenging activity or independent of scavenging activities. Please provide rationale supporting or data addressing either one (or both).

6. Provide the significance of the study. The LAB strains studied are not skin commensals. In lines 293 to 295, the authors claim that these strains can be used as probiotics for the anti-aging of skin. Please address how the unidentified factor(s) or LAB is implicated in the purpose. 

Author Response

Reviewer 2

Thanks for your attentive comments.

The manuscript was revised based on the comments in the review, with changes marked in red.

Below is a summary of the answers to each comment.

<GENERAL COMMENTS>

Point 1: It is not clear when and why UVB and CSF were treated. For antioxidant activity analysis, CSF was directly mixed with ABTS reagent. For ELISA assays, cells were first treated with UVB for 24 h and then treated with CSF for 24h (lines 103 to 104); whereas for general UVB treatments, cells were first treated with CFS for 24 h and then exposed to UVB (line 92 to 95). Please provide the rationale for treating UVB and CSF in different sequences in the text. It is presumed that cells were treated with CSF and then treated with UVB. It should be revealed that when qPCR and Western blot samples were taken after UVB treatment.

Response: Thanks for your comments. As noted by reviewers, we also pre-treated cells with CFS for 24 h and then exposed the cells to UVB (90-91). After UVB exposure, the cells were immediately replaced with fresh culture medium and treated with CFS for an additional 24 h for assay. Western blot samples were obtained within 48 h of UVB treatment. We modified the Method section (2.3-2.7) exactly and indicated the relevant reference [24].

Point 2: It is not clear why the radical scavenging activity of CSF is relevant to the subsequent experiments exploring the UVB protective effects of CSF. Strains of LAB were selected based on these activities for the following experiments, but CSF was not presented when UVB was exposed in these experiments. Please provide how the direct scavenging activity is involved in the UVB protective effects. Also, scavenging activity in the cytosol, where free radicals are made, is important. Please explain how direct scavenging effects in the extracellular are relevant in cells.

Response: Thanks for your comments. Reactive oxygen species (ROS) are generated by stimulation with UVB, and the production of pro-inflammatory cytokines is promoted to activate various signal transduction systems. This increases the generation of ROS, such as hydrogen peroxide, and reduces the expression of antioxidant enzymes. The increase in ROS changes the structure of genes and proteins, destroys cell homeostasis, and causes damage to the skin. Therefore, verification of antioxidant activity is very important.Recently, UVB has altered the gut microbiome by indirectly increasing serum vitamin D levels [1]. When ingested, probiotics can directly eliminate radicals in the intestine and increase the activity of antioxidant enzymes. In addition, the metabolites of probiotics are delivered to dermal cells through blood vessels and can provide various effects, such as reducing oxidative stress, supplying nutrients, and proliferating cells [2]. Therefore, in this study, treatment of LAB’s metabolites (CFS) on Hs68 cells (dermal fibroblast cell lines) exposed to UVB could confirm the indirect effect of probiotics on skin oxidative stress.

[1] De Pessemier B, Grine L, Debaere M, Maes A, Paetzold B, Callewaert C. 2021. Gut–skin axis: current knowledge of the interrelationship between microbial dysbiosis and skin conditions. Microorganisms. 9: 353.
[2] Boyajian JL, Ghebretatios M, Schaly S, Islam P, Prakash S. 2021. Microbiome and human aging: probiotic and prebiotic potentials in longevity, skin health, and cellular senescence. Nutrients. 13: 4550.

Point 3: Based on Fig. 1, CSFs positively affect cell proliferation. UVB will likely induce cell death which could be inhibited in CSF-treated cells. Since SOD1 and type I procollagen experiments in Fig. 1B and 1C were done in cell culture media, cell viabilities of CSF-treated cells in response to UVB exposure will likely affect the results. Therefore, cell viability assays of these cells should be provided.

Response: Thanks for your comments. We agree with you. According to the reviewer’s comment, we added the results confirming the cell protective ability of CFS in UVB-exposed Hs68 cells to Figure 1B. The results are described in lines 161-162.

Point 4: SOD1 is a cytosolic protein. Please provide the rationale for analyzing the levels in the cell culture media. Is it due to increased secretion or expression of these molecules? Please show if CSF induces mRNA or protein expression in Hs68 cells.

Response: Thank you for your comments. We agree with the reviewer. It may be more accurate to analyze SOD1 in cell lysates rather than CFS. Nevertheless, the expression of SOD1 in the cell culture supernatant could also be confirmed using the commercial ELISA kit, so no additional RNA or protein expression experiments were conducted. We plan to conduct further studies on skin oxidative stress in vivo animal model experiments.

Point 5: Fig 2-4 clearly show that CSF has anti-inflammatory and MAPK inhibitory effects. However, since CSF was not present when UVB was exposed, it is not clear if these effects are due to residual radial scavenging activity or independent of scavenging activities. Please provide rationale supporting or data addressing either one (or both).

Response: ROS causes an inflammatory response of skin cells directly. It mediates signal transduction processes and causes skin tissue damage involving inflammation or collagen degradation. In this study, CFS was pretreated for 24 h before UVB exposure, and CFS was further treated for 24 or 48 h after UVB exposure. Therefore, as suggested by the reviewer, it is thought that the radical scavenging activity and SOD1 expression by CFS treatment have a direct effect.

Point 6: Provide the significance of the study. The LAB strains studied are not skin commensals. In lines 293 to 295, the authors claim that these strains can be used as probiotics for the anti-aging of skin. Please address how the unidentified factor(s) or LAB is implicated in the purpose.

Response: Thanks for your comments. As noted by the reviewer, the strains used in this study are not skin commensal and do not directly affect the skin. However, as mentioned in point 2, intake of probiotics can improve gut microflora, and effective metabolites of these strains in the gut can flow to the skin through the bloodstream or affect the skin microbiome [31]. According to the reviewer’s comment, we have corrected the sentences in lines 249-251 and 302-204.
